# Belgian Consensus Recommendations to Prevent Vitamin K Deficiency Bleeding in the Term and Preterm Infant

**DOI:** 10.3390/nu13114109

**Published:** 2021-11-16

**Authors:** Simon Fiesack, Anne Smits, Maissa Rayyan, Karel Allegaert, Philippe Alliet, Wim Arts, An Bael, Luc Cornette, Ann De Guchtenaere, Nele De Mulder, Isabel George, Elisabeth Henrion, Kirsten Keiren, Nathalie Kreins, Marc Raes, Pierre Philippet, Bart Van Overmeire, Myriam Van Winckel, Vinciane Vlieghe, Yvan Vandenplas

**Affiliations:** 1Faculty of Medicine, KU Leuven, 3000 Leuven, Belgium; simon.fiesack@student.kuleuven.be (S.F.); kirsten.keiren@hotmail.com (K.K.); 2Department of Development and Regeneration, KU Leuven, 3000 Leuven, Belgium; anne.smits@uzleuven.be (A.S.); maissa.rayyan@uzleuven.be (M.R.); karel.allegaert@uzleuven.be (K.A.); 3Neonatal Intensive Care Unit, University Hospitals Leuven, 3000 Leuven, Belgium; 4Department of Pharmacy and Pharmaceutical Sciences, KU Leuven, 3000 Leuven, Belgium; 5Department of Clinical Pharmacy, Erasmus MC, 3011 Rotterdam, The Netherlands; 6Department of Paediatrics, Jessa Hospital, 3500 Hasselt, Belgium; dr.philippe.alliet@gmail.com (P.A.); fa303885@skynet.be (M.R.); 7Department of Paediatrics, ZOL Genk, 3600 Genk, Belgium; wim.arts@zol.be; 8Department of Pediatrics, ZNA Queen Paola Children’s Hospital, Faculty of Medicine UA, 2020 Antwerp, Belgium; anna.bael@zna.be; 9Department of Neonatology, AZ Sint-Jan, 8000 Brugge, Belgium; luc.cornette@azsintjan.be; 10Department of Paediatrics, Ghent University, 9000 Ghent, Belgium; dr.deguchtenaere@gmail.com; 11Vrije Universiteit Brussel (VUB), UZ Bussel, KidZ Health Castle, 1090 Brussels, Belgium; nele.demulder@uzbrussel.be; 12Neonatologie, AZ Groeninge, 8510 Kortrijk, Belgium; dr.isabelgeorge@gmail.com; 13Department of Neonatal Intensive Care, CHR Sambre et Meuse, 5000 Namur, Belgium; elisabeth.henrion@chrsm.be; 14Neonatal Intensive Care Unit, CHC MontLégia, 4000 Liège, Belgium; Nathalie.kreins@chc.be; 15Pediatric Department, CHC MontLégia, 4000 Liège, Belgium; pierre.philippet@chc.be; 16Kind en Gezin-Opgroeien, Vlaamse Overheid, 1000 Brussels, Belgium; bart.vanovermeire@opgroeien.be; 17Department of Paediatrics, Ghent University Hospital, 9000 Ghent, Belgium; Myriam.VanWinckel@uzgent.be; 18Neonatal Intensive Care Unit, Queen Fabiola Children’s University Hospital, Université Libre de Bruxelles, 1020 Bruxelles, Belgium; Vinciane.vlieghe@huderf.be

**Keywords:** vitamin K, vitamin K deficiency bleeding, term, preterm, prophylaxis

## Abstract

Neonatal vitamin K prophylaxis is essential to prevent vitamin K deficiency bleeding (VKDB) with a clear benefit compared to placebo. Various routes (intramuscular (IM), oral, intravenous (IV)) and dosing regimens were explored. A literature review was conducted to compare vitamin K regimens on VKDB incidence. Simultaneously, information on practices was collected from Belgian pediatric and neonatal departments. Based on the review and these practices, a consensus was developed and voted on by all co-authors and heads of pediatric departments. Today, practices vary. In line with literature, the advised prophylactic regimen is 1 or 2 mg IM vitamin K once at birth. In the case of parental refusal, healthcare providers should inform parents of the slightly inferior alternative (2 mg oral vitamin K at birth, followed by 1 or 2 mg oral weekly for 3 months when breastfed). We recommend 1 mg IM in preterm <32 weeks, and the same alternative in the case of parental refusal. When IM is perceived impossible in preterm <32 weeks, 0.5 mg IV once is recommended, with a single additional IM 1 mg dose when IV lipids are discontinued. This recommendation is a step towards harmonizing vitamin K prophylaxis in all newborns.

## 1. Introduction

Vitamin K (vit K) refers to a group of fat-soluble vitamins that are important in blood clotting, bone metabolism and regulating blood calcium levels. Neonates are born with very low vit K1 stores and vit K is barely detectable in cord plasma. Consequently, vit K prophylaxis is essential for all neonates [1]. Vit K does not easily cross the placenta, so that prenatal maternal vit K supplementation does not prevent vit K deficiency bleeding (VKDB). Townsend coined the term ‘hemorrhagic disease of the newborn’ in 1894 but it was not until the discovery of vit K (‘Koagulation vitamin’) by Dam and others in the 1930s that the pathophysiology was understood, allowing treatment and prophylaxis [2]. Vit K deficiency leads to a reduction in the activity of the Vit K-dependent coagulation factors II, VII, IX, X, and of the anticoagulant protein C and protein S. Proteins Induced by Vitamin K Absence (PIVKAs) have been used as surrogate and sensitive markers of vit K deficiency [3]. PIKVA-II is an immature form of prothrombin (absence of post-translational carboxylation), synthesized in the liver. In the absence of vitamin K or when its action is antagonized (warfarin), PIVKA-II is released into the blood.

Without prophylaxis, neonates are more prone to develop VKDB. VKDB is defined as ‘spontaneous bruising, bleeding or intracranial hemorrhage associated with prolonged clotting times but not due to an inherited coagulopathy or disseminated intravascular coagulation in neonates under 6 months of age’ [4]. VKDB is divided into three subcategories according to the age at onset: early VKDB occurring within 0 to 24 h after birth, classical VKDB between 1 and 7 days after birth and late VKDB after the first week of life [5]. Early VKDB is often located in the head (intracranial, cephalohematoma), intrathoracic, intra-abdominal or gastrointestinal tract [5]. Classic VKDB is mostly located in the gastrointestinal tract, umbilicus, skin, adrenal gland, nose or after circumcision. And late VKDB is often intracranial, skin, gastrointestinal. VKDB can be fatal or cause serious morbidity. In 50% of the late VKDBs, neonates present with intracranial hemorrhage [5]. Late vit K deficiency usually occurs in exclusively breastfed infants and or infants that have malabsorption disorders or hepatobiliary dysfunction, resulting in cholestasis and an impaired secretion of bile salts leading to malabsorption of vit K [5,6]. In the absence of prophylaxis, a VKDB incidence of 5 to 7 per 100,000 live births was reported in Europe [7]. 

A publication in the Lancet in 1944 initiated the worldwide practice of prophylactic administration of intramuscular (IM) vit K to neonates [8]. Preterm infants are potentially at greater risk for VKDB than term infants because of delayed enteral feeding and a subsequent delay in the colonization of their gastrointestinal tract with vit K producing microbiota, as well as immature hepatic and hemostatic function [9]. Frequent and long lasting antibiotic treatments may further affect microbiota growth [10]. However, the impact of the change in microbiota related to antibiotics regarding the risk of VKDB is not clear. The IM route became less obvious once Golding et al. reported in 1992 on the association between IM vit K prophylaxis and different types of childhood cancer [11]. However, despite multiple attempts to confirm this finding, the claimed increased risk in childhood cancer was never replicated [12,13,14]. Nevertheless, alternative routes of administration were investigated. The incidence of early and classical VKDB is also greatly reduced by administering vit K by either an oral or intravenous (IV) route [15,16]. As these alternative routes became part of our practices, parental refusal for IM prophylaxis became a major issue [17]. The pharmacokinetics of intravenous (IV) vit K are not very well known, but IV administration does not seem to bring the same efficiency as the IM route for the prevention of the late form of VKDB, especially if the injection is not repeated [18]. Moreover, anaphylactic allergic reaction to the IV injection of vit K has been described [19]. As a consequence, there is still a lot of debate and lack of consensus about the optimal regimen (dose, route) of vit K prophylaxis [20]. A clear, evidence-based guideline on optimal vit K prophylaxis is needed to provide the best medical care to all neonates. In this literature review, a comparison will be made between different regimens for term and preterm neonates, mainly focusing on prophylactic regimens to prevent late VKDB in breastfed neonates. Simultaneously, information on practices was collected from Belgian pediatric and neonatal departments. Based on the literature review and these practices, a consensus was developed and submitted for voting by the co-authors and the heads of the pediatric departments to reach a nationwide guideline in Belgium. 

## 2. Materials and Methods

The primary research question was: “What is the optimal dose and mode of administration for vit K prophylaxis in the preterm and term neonate?” A structured literature review was preformed to explore the best evidence-based regimen of vit K prophylaxis. Based on gestational age (GA), 3 groups were defined: very preterm infants with a GA < 32 weeks, moderate to late preterm neonates with GA 32 to 37 weeks and term neonates with GA ≥ 37 weeks. The intervention was defined as the prophylactic regimen (dose and route of administration) of vit K. The different regimens considered were: oral administration once at birth, oral administration 1x/day, oral administration 1x/week, oral administration multiple times in the first days/weeks of life, IM administration once at birth, IV administration once at birth, in different doses for the different routes of administration.

Early, classic, and late VKDB were considered as primary outcomes. Secondary outcomes were serum vit K level, PIVKA-II and prothrombin time (PT). The normal range in adults of serum vit K is 0.8–5.4 ng/mL [21]. A PIVKA-II level of 0.1 AU/mL or higher is considered as a cut-off for subclinical vit K deficiency in neonates. Since healthy adults do not develop VKDB, it is safe to assume that neonates with a PT equal to the adult range are less prone to develop VKDB. The range of PT in adults is 10.5–12.5 s [22].

The literature search was performed in February 2021 in the PubMed database using the mesh terms: (“Infant, Newborn”[Mesh] OR “Infant, Premature”[Mesh] OR “Infant, Extremely Premature”[Mesh]) OR (“Term Birth”[Mesh] OR “Premature Birth”[Mesh]) OR (“Infant, Low Birth Weight”[Mesh] OR “Infant, Very Low Birth Weight”[Mesh] OR “Infant, Extremely Low Birth Weight”[Mesh]) AND (“Vitamin K”[Mesh]). The articles were screened based on their title. Only articles about vit K prophylaxis in neonates were included for further screening of abstract and full text. In this final screening only clinical (investigational) trials and registry (observational) studies were included. Articles comparing at least 2 vit K regimens (oral, IM or IV) with well-defined dosages were included. Exclusion criteria were: other study types, other applications of vit K, no primary or secondary outcomes compatible with review outcomes and studies with a focus on professional or parental attitudes.

Alongside the structured review of the literature described above, a questionnaire on the current practice on vit K prophylaxis in Belgium was sent to all pediatric departments in Flanders and to all Belgian Neonatal Intensive Care Units (NICU). The responses are presented and discussed. All pediatric departments, NICUs and co-authors were offered to participate in a voting regarding the proposed administration of vit K, resulting in a broadly supported Belgian consensus recommendation. Statements for voting were scored numerically between 0 (strongly disagree) and 10 (strongly agree), with a predefined cut-off of 7 reflecting agreement (≥7) versus disagreement (<7). The Ethics Committee of the University Hospitals Leuven (Belgium) approved the questionnaire protocol (study MP017253). 

## 3. Results

### 3.1. Literature Review

A PRISMA flow chart was used to show the results of the literature search (Figure 1). 

The search delivered 1034 articles. After screening, 682 articles were retained. In the final review, 26 articles were included. Clinical data extracted from the included studies were vit K prophylaxis regimen with dose (in mg), GA of the neonates (in weeks), number of neonates included, incidence of VKDB, predefined outcomes (serum vit K at either time, PIVKA-II levels at either time, PT time at either time and/or incidence of VKDB), and number of affected neonates per outcome. An outcome was considered statistically significant when the *p* value was <0.05. 

Articles reporting at least two different prophylactic regimens of vit K in neonates were retained. Based upon the research question, 26 articles were retained in the final literature review, 11 registry studies and 15 clinical trials. An overview of these articles is provided in Table 1. 

Twenty-three of the articles cover healthy, full-term breastfed neonates. Three articles report on preterm neonates and two articles on neonates with biliary atresia. All but four articles had intramuscular (IM) administration once at birth as the reference regimen against other regimens. Overall, 675 articles were excluded (Figure 1). The main exclusion reasons were other study type (e.g., professional opinions and commentary articles on IM vit K prophylaxis), articles with no full text or other language (not Dutch or English), different methods of vit K prophylaxis (e.g., transdermal administration or administration of vit K prenatally to the mothers) and other aspects of prophylactic vitamin K (e.g., investigating adverse effects of vit K use). Finally, articles on the therapeutic use of vit K in neonates or young children with bleedings, rather than the prophylactic use, were excluded as well. 

In Table 2, a summary of the 15 included clinical trials is presented. The population evaluated in the papers is: 10 articles on GA ≥ 37 weeks, one article on GA ≥32- < 37 weeks, three articles with GA <32 weeks or low birth weight (mean birth weight of 1800 g) and one in which GA was not specified. IM administration once at birth is used in 13 articles (with doses varying from 0.2 to 5 mg), IV administration once at birth in three articles, oral administration once at birth in 12 articles, oral administration multiple times in the first days/weeks of life in two articles and no vit K prophylaxis as control in four papers. In two articles there was a comparison with formula-fed neonates. 

In Table 3, a summary of the 11 included registry studies is presented. In all articles the population consisted of neonates, not further specified, except for two papers on neonates with (not yet diagnosed) biliary atresia. The primary outcome of these publications is the incidence of VKDB in neonates. IM administration once at birth is used in nine articles, IV administration once at birth is not used, oral administration once at birth in two articles, oral administration multiple times in the first days/weeks of life in seven papers and no vit K prophylaxis as control in four. In one article there was a comparison between breast- and formula-fed neonates. In three articles the dose administered was not specified. 

A detailed representation of the results of the registry studies is presented in Table 2. The incidence of VKDB in neonates receiving vit K 1 mg IM once at birth was statistically significantly lower or equal to other regimens. Based on the mean incidence of the different regimens, 1 mg IM once at birth, 1 mg or 2 mg oral administration once at birth followed by 1 mg oral once per week, and formula-fed neonates have a significantly lower incidence compared to other regimens. It is noted that the weekly oral regimen was only reported once, leading to potential bias.

Three articles discussed all predefined secondary outcomes of this review [22,24,30]. Three articles reported short term variables within the first 24 h of life [31,32,34], 10 articles between day 1 and day 7 of life [16,22,26,27,28,30,31,46,47,48] and eight articles after the first week of life [22,24,26,27,28,30,32,34], correlating with early, classic and late VKDB, respectively. 

In term neonates the levels of serum vit K in the first 24 h when given 1 mg IV or 1 mg IM once at birth was significantly higher compared to 1 or 2 mg oral once at birth. The serum vit K level in the first week of life of the term neonate is not different between the 2 mg oral once at birth and the 1 mg IM once at birth regimen. Schubiger et al. reported in 1993 that a significantly higher serum vit K level at day 4 of life was observed when 3 mg oral once at birth was administered compared to 1.5 mg IM once at birth [34]. 

After the first week of life, a significantly higher serum vit K level is documented in the groups receiving 1 mg or 1.5 mg IM once at birth compared to 1 mg or 3 mg oral once at birth [22,27,28]. Greer et al. reported a significantly higher serum vit K level when 3 × 2 mg oral in the first days of life was administered compared to 1 mg IM once at birth [30] and Cornelissen et al. reported significantly higher serum vit K levels in the weekly or daily oral regimen compared to either IM or PO once at birth [28].

The articles on premature or low birth weight neonates (mean birth weight 1800 g) as population, compared different vit K dosages for the same route of administration [24,28,32]. No significantly different serum vit K level was reported when the same dose was administered, independent of the route of administration. In the first 24 h, the first week and up to the postnatal age of 25 days the reported serum vit K in all articles were equal to or above the adult range (0.8–5.4 ng/mL).

Only one article documented significantly different PIVKA-II levels. In this article, the PIVKA-II levels were significantly higher at 12 weeks of life after 1 mg oral or IM once at birth compared to the weekly or daily oral regimen of 25 µg in term neonates [28].

PT was not significantly different between the investigated regimens, except for one paper by O’Connor et al. 1986, reporting a significantly higher PT when no vit K prophylaxis was given compared to 2 mg oral or 1 mg IM once at birth [16]. In the other studies, PT values were within or above the adult range in the first days of life [22,24,25,30,35].

### 3.2. Current Practices in Belgium

Table 3 and Table 4 summarize the daily practice in the maternity wards in Flanders (response rate 95% (54/57)) for formula and breastfed term born infants [49]. The ratio IM versus oral administration is close to a fifty-fifty distribution. In 9 out of 13 maternity wards (17% of all participating maternity wards) that adapted the policy regarding vit K prophylaxis after 2015, the change was from oral towards IM prophylaxis. The other adaptations concerned dosage. Table 5 lists the currently used vit K prophylaxis protocols in the Belgian Neonatal Intensive Care Units (NICU).

**Table 3 nutrients-13-04109-t003:** Vitamin K prophylaxis for term breastfed infants in Flanders (Reprinted with permission from ref. [49]. Copyright 2021 Belgian Society of Paediatrics.).

Route	N° Responses	Regimen	N° Responses (%)	
Intramuscular	29 (54%)	1 mg IM immediately after birth	28	(52%)
		2 mg IM immediately after birth	1	(2%)
Oral	25 (46%)	2 mg oral at birth and maintenance dose 1–2 mg/week	14	(26%)
		1 mg oral at birth and maintenance dose 150 µg/day	6	(11%)
		2 mg oral at birth and maintenance dose 25 µg/day	3	(6%)
		2 mg oral at birth and maintenance dose 150 µg/day	1	(2%)
		2 mg oral at birth: no information on maintenance dose	1	(2%)
		2 mg oral at birth, 2 mg oral day 4–6 and 2 mg at 4–6 weeks	0	(0%)

### 3.3. The Consensus

Based on the literature review, the following statements were sent to all co-authors and to the heads of the departments of pediatrics in Flanders. The response rate was 56/74 (75.7%) (some of the co-authors are also heads of departments). All statements reached consensus (84–100%) (Table 6). The major reason for disagreement was restraint from IM injection in preterms, despite the minimal volume (0.1 mL for 1 mg).

## 4. Discussion

This review of the literature allows to conclude that considering advantages and disadvantages of each regimen, 1 mg IM vit K once at birth to all term neonates can be advised as a first option. Because most infant formula is fortified with approximately 50 µg/L of vit K (range from 39 µg/L in starter formula to 85 µg/L in human milk fortifier, compared to 3 µg/L in human milk), VKDB is now mostly confined to breastfed neonates [20]. 

The prevention of VKDB was introduced in the 1940s with the administration of vit K [8]. Over the last decades, an international debate about the best prophylactic regimen was ongoing and still remains without consensus [50]. The pros and cons of oral versus IM prophylaxis, as well as the most optimal dose, is still a topic of discussion [20]. A single dose of vit K orally at birth prevents early and classical VKDB; however, this regimen fails to prevent late VKDB, even in very high dosages [51]. A single dose of vit K IM at birth (dose not mentioned) is reported to be the most effective in preventing classical as well as late VKDB, reducing the incidence to 0.2 per 100,000 live births [7]. Despite a long experience with prophylactic vit K, failures of adherence to prophylactic oral administration regimens still occur. Parents may simply forget to administer the oral vit K. These failures also occur in neonates who were later diagnosed with a cholestatic liver disease or other diseases related to malabsorption such as cystic fibrosis [52,53]. Due to the absence of intestinal bile in cholestatic neonates, the absorption of vit K and other fat-soluble vitamins is greatly reduced. 

The overall incidence of VKDB decreased over time since an increasing number of countries and maternities implemented more effective regimens of vit K prophylaxis. Since 1961 the American Academy of Pediatrics (AAP) has recommended a single dose of 0.5 or 1 mg IM vit K once at birth or 1 or 2 mg oral vit K 1x at birth to the neonate to prevent VKDB [54]. The 2003 recommendation by the AAP is 1 mg IM vit K once at birth [50,55]. The Canadian Paediatric Society recommended a similar prophylaxis regimen, but suggested as an alternative a regimen of 2 mg oral within 6 h after birth, repeating it at 2 to 4 weeks and at 6 to 8 weeks of age [56]. In 2018, an oral regimen was added to the Canadian recommendations: 2 mg oral vit K at birth following a weekly dose of 1 mg oral vit K for 3 months [57]. The European Society for Paediatric Gastroenterology Hepatology and Nutrition (ESPGHAN) suggests the same regimens as the Canadian recommendation [58]. Wariyar et al. reviewed the incidence of late VKDB in various population studies, excluding cases where bleeding occurred because prophylactic vit K was not given, and could confirm the need for vit K prophylactic administration to prevent late VKDB [59]. It is strongly suggested in this paper that IM administration is more effective (Table 7) [59]. 

In 2016, Witt et al. compared the efficacy of the Dutch regimen of oral prophylaxis (1 mg of vit K orally at birth followed by 150 μg daily from week 2 to week 13) with the Danish regimen (single IM dose of 1 mg at birth) in breastfed infants [44]. It has been reported that an error occurred in the original paper, as the IM dose was 1 mg, not 2 mg [44]. Results evidenced that oral strategy failed to prevent VKDB in infants with undiagnosed cholestasis, such as biliary atresia [44]. Prevention was effective in the case of IM administration. However, even after administering 1 mg IM once at birth or 1 mg oral weekly, not all cases of late VKDB were prevented [41,60,61]. Table 8 provides information of recommendations for vit K prophylaxis in term infants for different countries. Different forms of vit K exist: phylloquinone and menaquinones [60]. Phylloquinone, which is the major dietary source, is concentrated in leafy plants and is the vitamin K form best characterized in terms of food composition and dietary intake. In contrast, menaquinones are the product of bacterial production or conversion from dietary phylloquinone. Most countries recommend phylloquinone for vit K prophylaxis, except Japan, which uses menaquinone-4 [60].

The questionnaire to retrieve information regarding the daily practice for vit administration in term born infants in the maternity wards in Flanders had a remarkable response rate of 95% (54/57) [49]. The outcome of this questionnaire illustrates very well the debate about the best way to administer vit K prophylaxis. While there is 100% conviction that vit K administration immediately after birth is mandatory, there is no agreement about the best way to administer the vit K, since 54% administer vit K IM and 46% oral [49]. When the same colleagues were contacted to vote on the new consensus statements based on this review of the literature, we succeeded in convincing most of the respondents to switch to IM administration since only 5 % continued to prefer the oral administration of vit K. 

Compliance is not an issue when parenteral prophylaxis is applied, except if the administration is unintentionally omitted. Compliance is a more relevant issue in the case of oral prophylaxis. Table 9 summarizes the pros and cons of IM versus oral prophylactic administration of vit K. Studies showed an increased prevalence and incidence of late VKDB in the event of oral prophylaxis [37,62]. Zurynski et al. reported in 2020 a significant increase in parental refusal: in the second half of the study, the number of VKDB cases following parental refusal doubled [43]. Busfield et al. stated parental refusal already in 2013 as the most important issue in the prophylactic prevention of VKDB [41]. Loyal et al. reported that the refusal of IM vit K by parents was notably higher among home births and in birthing centers. The main reasons for refusal were: concern of pain and harm from the injection and a desire to be natural and a belief in alternative methods of prophylaxis [63]. There are many factors contributing to determine an optimal vit K administration regimen e.g., cost, accessibility, compliance, ease of use, effectiveness, contra-indication, side-effects, parental refusal, or health care organizations. A risk with an IM injection is a potential local reaction such as an infection or muscle bleeding at the injection site. Von Kries investigated in 1992 potential complications following IM injection of vit K and found no significant complications in a study population of 420,000 neonates [36]. 

Pain as a side-effect of an IM injection, and prevention of pain and discomfort are reasons for a higher parental refusal [34]. We are aware that our assessment of the literature and survey has focused on VKDB and does not cover the topic of pain management, but meta-analytical evidence on effective management of this needle and injection pain has been published, as Cochrane reviews highlight the efficacy in pain relief of skin-to skin contact, mother’s milk and sucrose during an IM injection [64,65,66]. In essence, such types of interventions include swaddling/containment, skin-to-skin care, breastfeeding or sweet tasting solutions to stimulate sucking. Effective implementation of such strategies is relevant beyond the emotional and ethical aspects of pain management, as knowledge and awareness of the (long-term) psychological effects of pain in neonates increases [67]. There is an obvious need to minimalize pain during treatment procedures, including vit K prophylaxis. 

Literature on IV administration of Vit K is very limited. However, anaphylactic reactions have been reported [19]. IV administration may be considered as an effective way to administer Vit K in newborns who already have an IV access, quite common, e.g., in preterm and (critically) ill neonates. Despite the limited data available, the majority of the Belgian NICUs prefer the IV route for vit K administration. Furthermore, the current recommendations for vit K prophylaxis for premature neonates vary widely. 

Preterm neonates are at a higher risk of developing VKDB, due to hepatic immaturity and delayed and less-diversified gut colonization [66]. Bacterially-produced menaquinones, 2-methyl-1,4-naphthoquinones with an unsaturated polyisoprenoid chain at the 3-position, are biologically active forms of vit K that are present in high concentrations in the human lower bowel [46]. Menaquinones may only partially satisfy the human requirement but their contribution seems much less than previously thought [46]. While literature discusses “preterm” versus “term”, recommendations focus only on birthweight, choosing 1500 g as the birthweight cut-off. Therefore, whenever preterm in the context of vit K prophylaxis is mentioned, a birthweight <1500 g is intended. Clarke et al. suggested 0.2 mg IM once at birth for premature neonates for the prevention of early and classic VKDB [24]. The authors claim that a higher dosage could lead to an accumulation of vit K, but they mentioned as well that three infants from the 0.2 mg group had undetectable serum vit K1 as early as the third postnatal week without any evidence of even a mild functional deficiency [60]. On the other hand, adverse effects of a high-dosage administration of vit K were not reported. A Cochrane review from 2018 concluded that dosage studies suggest that doses of vit K administered to preterm infants lead to supraphysiologic levels [47]. The authors of the Cochrane review conclude that in the absence of evidence that vit K is harmful or ineffective and since vit K is an inexpensive drug, it seems prudent to follow the recommendations of expert bodies and give vit K to preterm infants [47]. The voting among Belgian pediatrics illustrates the restraints for IM injection in preterm, despite the small volume of 0.1 mL, and despite the poor knowledge about the metabolism after IV injection and possible anaphylactic reactions [19]. Further research is needed to establish an optimal dose in preterm. In the meantime, we recommend the same attitude independent of the gestational age. When IM administration is not possible, a single administration of 0.5 mg IV is recommended. Since IV administration of lipids is part of total parenteral nutrition and since most IV lipids do contain vit K, repeat administrations of vit K are not recommended as long as IV lipids with vit K are administered. When IV feeding is discontinued, IM administration of 1 mg vit K is recommended. 

## 5. Conclusions

It has been reported to be ethical to provide nudges in medical decision-making about vit K prophylaxis [48]. During prenatal visits, the injection should be reviewed, discussed, recommended as the default and reinforced that this is the best way to keep their upcoming newborn baby healthy [48]. Prophylactic vit K supplementation is necessary to prevent VKDB in neonates and infants. However, the route of administration and most optimal dosage are still debated. Considering evidence form literature and recommendations by scientific societies and by multiple countries, a consensus was reached by the authors of this paper resulting in a Belgian recommendation.
For term born infants, we recommend the administration of 1 mg IM vit K once at birth to all term neonates.In case of parental refusal of the IM administration, healthcare providers should inform the parents about a slightly inferior alternative, 2 mg oral vit K at birth followed by the administration of 1 or 2 mg oral vit K, weekly for 3 months in breastfed infants, with specific attention to compliance. No further supplementation is needed after birth in formula-fed infants, neither in mixed breast nor formula feeding once formula feeding exceeds 50% of the intake.In preterm neonates (also <32 weeks’ gestation), we recommend the same approach as in term neonates. In case IM administration is not possible, 0.5 mg IV single administration is recommended, followed by a 1 mg IM administration when intravenous lipids are discontinued.Infants with cholestasis or another disease associated with fat-malabsorption need vit K (and other fat-soluble vitamins) supplementation regardless of the mode of feeding in order to prevent vitamin K deficiency-related coagulation disorders.

## Figures and Tables

**Figure 1 nutrients-13-04109-f001:**
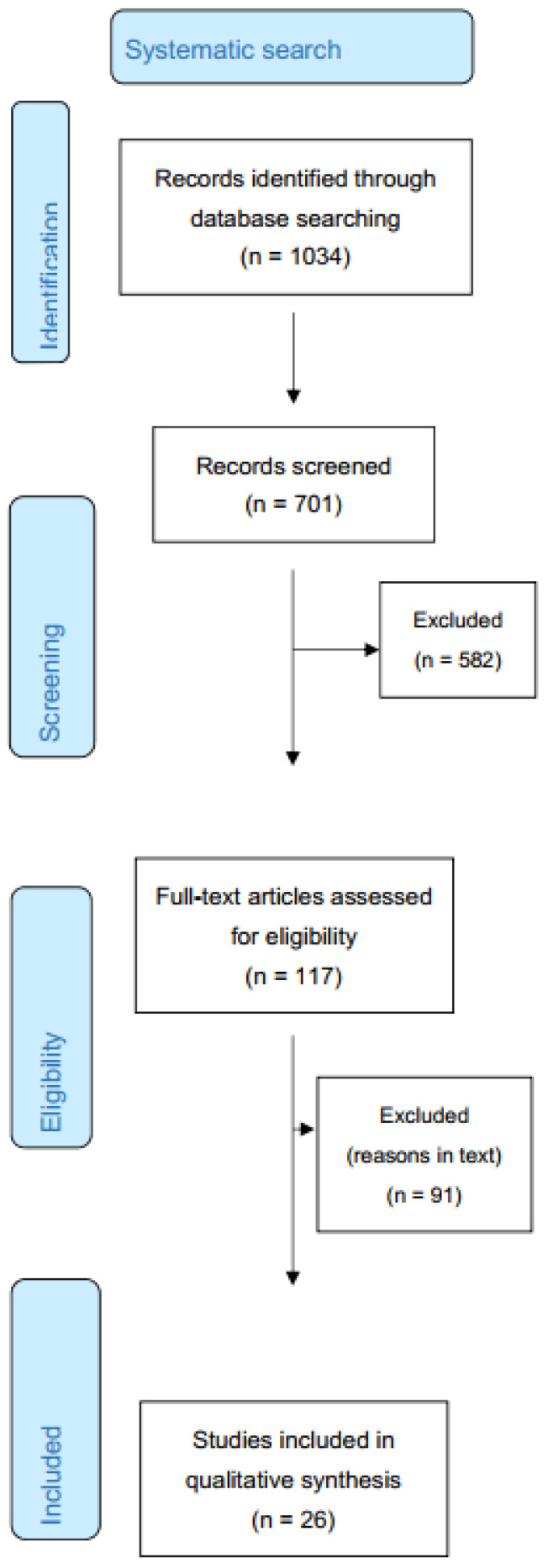
Flow chart literature search.

**Table 1 nutrients-13-04109-t001:** Summary of included clinical trials in the literature review.

References	Vitamin K Prophylaxis Regimens (mg)	GA (Weeks) Birthweight	N° Study Subjectis
Costakos 2003 [23]	IM or IV not specified (1 vs. 0.5)	<32	27
Clarke 2006 [24]	IM at birth vs. IM at birth vs. IV at birth (0.5 vs. 0.2 vs. 0.2)	<32	90
Jørgensen 1991 [25]	1 mg PO at birth vs. 1 mg IM at birth	≥35	300
O’Connor 1986 [16]	none vs. 2 mg PO at birth vs. 1 mg IM at birth	≥37	60
Hathaway 1991 [26]	2 mg PO at birth vs. 5 mg PO at birth vs. 1 mg PO at birth vs. none	≥37	36
Cornelissen 1992 [27]	1 mg PO at birth vs. 1 mg IM at birth	≥37	331
Hogenbirk 1993 [22]	Formula vs. none vs. 1 mg PO at birth vs. 1 mg IM at birth	≥37	80
Cornelissen 1993 [28]	1 mg PO at birth vs. 1 mg IM at birth vs. 1 mg PO 1×/week vs. 0.025 mg PO 1×/day	≥37	447
Gupta 1994 [29]	1 mg IM at birth vs. 2 mg PO at birth	≥37	176
Greer 1998 [30]	1 mg IM at birth vs. 3×2 mg PO	≥37	134
Pereira 2003 [31]	2 mg PO at birth vs. 1 mg IV at birth	≥37	44
Sann 1985 [32]	none vs. 2 or 5 mg PO or IM at birth not specified	BW (mean) 1800 g	26
McNinch 1985 [33]	1 mg IM at birth vs. 1 mg PO at birth vs. 1 mg PO with first feed	BW > 2000 g	107
Schubiger 1993 [34]	1.5 IM at birth vs. 3 mg PO at birth	BW > 2000 g	25
Shoshkes 1959 [35]	1 mg IM at birth vs. 2 mg PO at birth vs. none	?	91

Legend: GA: Gestational Age; IM: intramuscular; IV: intravenous; PO: per os (oral); BW: birthweight; N°: number; ?: not known.

**Table 2 nutrients-13-04109-t002:** Summary of included registry studies in the literature review.

Reference	Vitamin K Prophylaxis	Period	N° of VKDB/Population (%)
von Kries 1992 [36]	none vs. 1 mg IM at birth vs. 2 mg PO at birth	1988–1989	14/750,000 (0.0019)
Cornelissen 1997 [37]	1 mg PO at birth + 0.025 mg/day vs. 3 × 1 mgPO vs. 1 mg IM at birth vs. 2 × 2 mg P0	1992–1995	49/2,372,000 (0.0021)
von Kries 1999 [4]	3 mg PO vs. 3 × 1 mg PO	1995–1998	23/3,200, 000 (0.007)
Hansen 2003 [38]	2 mg PO at birth + 1 mg/week vs. 1 mg IM at birth + 1 mg PO 1×/week	1992–2000	0/ 507,850 (0)
McNinch 2007 [39]	none vs. 1× PO, 2× PO or 3× PO in first days vs. IM at birth	1988–1990 1993–1994 2001–2002	27/1,671, 000 (0.016) 32/1,609,785 (0.020) 7/1,456,200 (0.005)
Darlow 2011 [40]	none vs. IM at birth (not specified)	1998–2008	17/1,288,018 (*)(0.013)
Busfield 2013 [41]	none vs. IM at birth vs. PO at birth (not specified)	2006–2008	11/1,700,000 (0.007)
Löwensteyn 2019 [42]	1 mg PO at birth + 0.025 /day vs. 1 mg PO at birth + 0.150 mg/day Late intracranical VKDB	2008–2011 2011–2015	18/583,117 (0.031) 10/843,820 (0.011)
Zurynski 2020 [43]	3 × 2 mg PO vs. 1 mg IM once at birth	1993–2017	58/6,904,762 (*)(0.008)
Witt 2016 [44] °	1 mg PO at birth + 0.025 mg/day vs. 1 mg PO at birth + 0.150 mg/day vs. 2 mg IM at birth (biliary atresia)		55/90 (611)
van Hasselt 2008 [45]°	1 mg PO at birth + 0.025/day vs. 2 mg PO at birth + 1 mg/week vs. 2 mg IM at birth vs. none in Formula (biliary atresia)		28/151 (185)

Legend: VKDB = vitamin K deficiency bleeding, PO = per os, IM = intramuscular; IV: intravenous; (*) = calculated based upon overall incidence.; °: both studies are in infants with biliary atresia; the cholestasis causes malabsorption of lipids and thus also of lipid soluble vitamins.

**Table 4 nutrients-13-04109-t004:** Vitamin K prophylaxis for term formula-fed infants in Flanders (Reprinted with permission from ref. [49]. Copyright 2021 Belgian Society of Paediatrics.).

Route	N ° Responses	Regimen	N° Responses(%)	
Intramuscular	30 (56%)	1 mg IM at birth	29	(54%)
		2 mg IM at birth	1	(2%)
Oral	24 (44%)	2 mg oral at birth	23	(43%)
		1 mg oral at birth	1	(2%)

**Table 5 nutrients-13-04109-t005:** Vitamin K administration immediately at birth in NICUs in Belgium (preterm and sick term).

Centre	Preference	Route	Follow-Up
1	<1 kg	IV 1 mg oral 10 µg (2 dr)°	None ?
	1–2 kg	IV 1 mg oral 2 mg oral 10 µg (2 dr)°	None ? 25 µg/day up to 3 months if breastfed and no HMF
	>2 kg	IV 2 mg oral 2 mg oral 25 µg (5 dr)°	None ? 25 µg/day up to 3 months if breastfed and no HMF
2	<39 weeks and/or < 3000 g	IV 2 mg	If IV lipid < 2 g/kg/day: 1 mg vit K/day IV If > 50% breastmilk: 10 µg/day (2 drops)
3	<1500 g	IV 0.5 mg	0.5 mg IV/day (up to stop infusion)
	>1500 g	IM 1 mg	
4	(partial)(total) PN		>36 weeks: IV 1 mg/day <36 weeks: IV 1 mg/day up to day 6, followed by IV 1 mg 3x/week
	Breastfeeding (>50% intake)		In NICU: oral 2 mg/week up to 3 months Out: oral 25 µg/day (5 dr)
5	<34 weeks	IV 1 mg	oral or IV 1 mg/week during 10 weeks
	≥34 weeks	IV 2 mg	oral or IV 2 mg/week during 10 weeks
6	Preterm in NICU	IV 1 mg	If TPN: no vit K If breastfeeding with HMF/formula: no vit K If breastfeeding: 25 µg/day (5 dr) when dismissed
7	<36 wks	IM 0.5 mg	None
		IV 0.5 mg	oral 1 mg /week up to diversification
8	All ages	IV/IM 1 mg	
9	<35 weeks and <1500 g	IV 0.5 mg	>Day 7 and no IV line: oral 2 mg/week if breastfeeding
	<35 weeks and ≥1500 g	IV 1 mg	>Day 7 and no IV line: oral 2 mg/week if breastfeeding
	>35 weeks and NICU	IV 1 mg	>Day 7 and no IV line: oral 2 mg/week if breastfeeding
	>35 weeks and healthy	PO 2 mg	>Day 7 and no IV line: oral 2 mg/week if breastfeeding
10	>35 weeks and NICU	IV/IM 2 mg	None
11	≥36 weeks <36 weeks	IM 1 mg IV 1 mg/kg	None Day 1–14: IV 0.1 mg/kg (minimum 0.1 mg), or oral 1 mg (<2 kg) to 2 mg (>2 kg). Day 14–3 months if breastfeeding: same regimen 1x/week

Legend: °: if > 50% mother’s milk and no human milk fortifier or vitalipid; NICU: neonatal intensive care unit, HMF: human milk fortifier; TPN: total parenteral nutrition.

**Table 6 nutrients-13-04109-t006:** Voting results (scores 0–10) on the statements on vitamin K prophylaxis.

Statements	Mean	Disagree (N) (Score)
Statement 1. For term born infants, we recommend the administration of 1 mg IM vit K once at birth to all term neonates.	9.02	5/56 (2,4,5,5)
Statement 2A. In case of parental refusal of the IM administration, the health care provider should inform the parents about a slightly inferior alternative option: 2 mg oral vit K at birth followed by the administration of 1 or 2 mg oral vit K 1x per week during 3 months in breastfed infants, with specific attention to compliance.	9.07	3/56 (2,4,6)
Statement 2B. No further supplementation is needed after birth in formula-fed infants, even in mixed breast- and formula-feeding (if formula feeding > 50% of the intake).	8.75	5/56 (1,4,5,6,6)
Statement 3A. In preterm neonates (also <32 weeks gestation), we recommend the same approach as in term neonates.	8.41	4/56 (1,1,3,4)
Statement 3B. In case IM administration is not possible, 0.5 mg IV single administration at birth is recommended, followed by 1 mg IM administration when IV lipids are discontinued.	8.25	9/56 (1,1,4,5,5,6,6,6,6)
Statement 4. Infants with cholestasis or another disease associated with fat-malabsorption need vit K (and other fat-soluble vitamin) supplementation regardless of the mode of feeding in order to prevent vitamin K deficient coagulation disorder.	9.39	0

Legend: N: number of votes.

**Table 7 nutrients-13-04109-t007:** Number (/10,000 cases) of late vitamin K deficiency bleeding in different populations according to prophylactic regimen (Adapted from ref. [59]).

Prophylaxis	Country (Period)	N° of Late VKDB/10,000
No	Japan 1978–1980	860
	Japan 1981–1985	718
	UK 1988–1989	454
	Germany 1988	721
1–2 mg oral once at birth	Switzerland 1986–1987	642
	UK 1988–1989	1420
	Sweden 1987–1989	511
	Denmark 1990–1992	446
1 mg oral at birth and at 1 and 3–5 weeks	Australia 1993–1998	197
	Germany 1993–1994	183
2 mg oral at birth and at 1 and 4 days	Switzerland 1995	121
1 mg oral at birth and at 2, 4 and 6 weeks	North of England 1993–1998	103
1 mg oral at birth and 25 µg/day up to 3 months	Netherlands	68
2 mg oral at birth and 1 mg weekly for 3 months	Denmark 1993–1998	0
1 mg intramuscular at birth	UK 1988–1989	0
	Australia 1993–98	0.01

Legend: Number (N°) of late VKDB: number of late vitamin K deficiency bleedings.

**Table 8 nutrients-13-04109-t008:** Vitamin K recommendation to prevent VKDB in term infants.

Country	Preference	Route at Birth	Follow-Up
Germany	IM/oral	IM 1 mg oral 2 mg	None 2 mg week 1 and week 4
Austria	Oral	oral 2 mg	2 mg day 4–6 and 2 mg week 4–6
UK	IM	IM 1 mg oral 2 mg	None 2 mg week 1 and 2 mg week 4 (only BF)
Ireland	IM	IM 0.8–1.0 mg ° oral 2 mg	None 2 mg day 4–7 and 2 mg week 4 (only BF)
Danmark	IM	IM 2 mg	None
Norway	IM	IM 0.5–1 mg	None
Finland	IM	IM 1 mg oral 2 mg	None 2 mg week 1 and 2 mg week 4
France	oral	oral 2 mg	2 mg day 3–7 and 2 mg week 4
Greece	IM	IM 1 mg	None
Italy	IM	IM 2 mg	None
Latvia	IM/oral	IM 1 mg oral 2 mg	None 2 mg week 1 and 2 mg week 3–4
Lithuania	IM	IM 1 mg (>1500 g BW) oral 2 mg	None 2 mg day 3–7 and 2 mg week 6
Czech Republic	IM/oral	IM 1 mg oral 2 mg	None 2 mg/week during 10–12 weeks
Slovenia	IM	IM 1 mg	None
Switzerland	oral	oral 2 mg	2 mg day 4 and 2 mg week 4
USA	IM	IM 0.5–1 mg oral 2–4 mg after 1st feeding	None 2 mg week 2–4 and 2 mg week 6–8 2 mg week 1 and 2 mg/week during BF 2 mg week 1 and 25 µg/day during 13 weeks
Canada	IM	IM 1 mg (≥1500 BW)	None
Australia New Zealand	IM	IM 1 mg (≥1500 g) oral 2 mg IV0.3 mg/kg (0.2–0.5 mg/kg) *	None 2 mg day 3–5 and 2 mg week 4–6 (day 22–28) Repeat if vomits within 1 h or diarrhea within 1 day after administration Can be repeated weekly
WHO	IM	IM 1 mg	None
ESPGHAN	IM	IM 1 mg oral 2 mg	None 2 mg day 4–6 and 2 mg week 4–6 1 mg/week for 3 months

Legend: IM: intramuscular; BF: breastfeeding; °: depending on weight; *: in sick infants if unable to give IM or oral; WHO: World Health Organization; ESPGHAN: European Society for Paediatric Gastroenterology Hepatology and Nutrition.

**Table 9 nutrients-13-04109-t009:** Pros and cons of intramuscular and oral prophylactic administration of vitamin K.

Intramuscular Vitamin K	Oral Vitamin K
Advantages	Advantages
Easy to use Highest efficacy Optimal compliance Minimal cost	Easy to use Less parental refusal
Disadvantages	Disadvantages
Pain and discomfort, but can be managed effectively [36,63,64] Potential local reaction, but uncommon [43] Parental refusal	Lower efficacy Compliance/adherence Less effective in neonates with biliary atresia

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
