# Peer review of "Belgian Consensus Recommendations to Prevent Vitamin K Deficiency Bleeding in the Term and Preterm Infant"

_nutrients, 2021, doi:10.3390/nu13114109_

Round 1

Reviewer 1 Report

Dear Authors, thank you for submitting this comprehensive review of VKBD.

My comments:

Title: better 'recommendations'

Abstract: does not mention survey methods and results in detail.

Introduction: should mention possible allergic reaction to intravenous vitamin K.

Results:

Table 2: 4th column - 'Number of study subjects'.

Table 3: omit 5th column - should be written in brackets and added to 4th column. References Witt and van Hassell give figures that are out of proportion with the other figures in that table (611 and 185). They should be reviewed and omitted or explained in more detail in legend.

Under 3.2 you cite results of another publication (ref 45) which is apparently in press at another journal. This could could be construed as duplicate publication. You should submit the proofs of ref 45 to the Editor of Nutrients.

Discussion: omit 'exhaustive' and Table 8. This table is an incomplete selection of formula milks. The information can be provided in one sentence. Page 12 bottom: 'daily practice'. Page 13 top: 'convincing'. In the discussion a paper on 'nudging' should be mentioned (Moses, Borecky, Dubov. It's ok to nudge for vitamin K. Acta Paediatrica 2019).

Author contributions: they are not detailed enough.

References: they should be re-checked (there are several formatting issues, e.g. 7, 22-26, 61.

Author Response

Reviewer 1

Title: better 'recommendations'

Done

Abstract: does not mention survey methods and results in detail.

We agree with the reviewer, but the maximal number of words is 200. We have now 194 words. And we think it is not further possible to decrease this in order to add the survey methods and results in detail. Obviously, this information is provided in the full version of the paper.

Introduction: should mention possible allergic reaction to intravenous vitamin K.

We thank the reviewer for this additional information. It was added to the introduction and discussion section, with a reference.

Table 2: 4th column - 'Number of study subjects'

Done

Table 3: omit 5th column - should be written in brackets and added to 4th column

Done

References Witt and van Hassell give figures that are out of proportion with the other figures in that table (611 and 185). They should be reviewed and omitted or explained in more detail in legend.

We did choose to mention the reason in the legend (which was already mentioned in column 2)

Under 3.2 you cite results of another publication (ref 45) which is apparently in press at another journal. This could could be construed as duplicate publication. You should submit the proofs of ref 45 to the Editor of Nutrients.

We did submit the proofs of the reference to the Editor.

Discussion: omit 'exhaustive' and Table 8. This table is an incomplete selection of formula milks. The information can be provided in one sentence.

We understand this remark of the reviewer and adapted the manuscript

Page 12 bottom: 'daily practice'.

corrected

Page 13 top: 'convincing'

corrected

In the discussion a paper on 'nudging' should be mentioned (Moses, Borecky, Dubov. It's ok to nudge for vitamin K. Acta Paediatrica 2019).

We thank the reviewer for the suggestion. We did add the reference (conclusion).

Author contributions: they are not detailed enough.

Author Contributions: Conceptualization, S.F., A.S., K.A. and Y.V.; methodology, S.F., A.S., K.A., M.v.W., K.K. and Y.V.; data curation, S.F., A.S., K.A., M.v.W. and K.K.; writing—original draft preparation, S.F., A.S., M.R., K.A. and Y.V; writing—review and editing, S.F., A.S., M.R., K.A., P.A., W.A., P.A., W.A., A.B., L.C., A.d.G., N.d.M., I.G., E.H., K.K., N.K.; M.R.; P.P.; B.v.O.; M.v.W., V.V. and Y.V.; supervision, A.S., K.A., M.v.W., and Y.V. All authors have read and agreed to the published version of the manuscript. Please turn to the CRediT taxonomy for the term explanation.  

References: they should be re-checked (there are several formatting issues, e.g. 7, 22-26, 61.

done

Reviewer 2 Report

This is a well-written review of literature about vitamin K prophylaxis and vitamin K deficiency bleeding (VKDB) in newborns, which was utilized to inform standard practice in Belgium.  Survey results about current practices in a maternity ward in Flanders are also included. The authors advised that 1-2 mg given intramuscularly (IM) to term infants and 1 mg IM to pre-term infants (<32 weeks). If parents refuse, 2 mg is recommended orally at birth followed by 1-2 mg/week for 3 months for breast fed infants. These advised amounts are consistent with the scientific literature and practices in other countries.

A few comments:

The authors might consider indicating that multiple forms of vitamin K exist (phylloquinone and menaquinones), how they differ, and that most countries recommend phylloquinone for vitamin K prophylaxis, except Japan which uses menaquinone-4 (Shearer M, Blood, March 2009, Pages 49-59). Another option would be to include "vitamin K form' as a column in Table 10.

The authors indicate that lack of gut bacteria in newborns contributes to VKDB (lines 86, 362). However, vitamin K produced by intestinal bacteria is not well-absorbed so its contribution to preventing VKDB is questionable (10.1146/annurev.nu.15.070195.002151). Likewise, while antibiotics can alter the intestinal bacteria, it is uncertain how this affects VKDB (lines 87-88).

Lines 235-237: It would be helpful to include references so that the reader knows which studies these are.

Author Response

Reviewer 2

This is a well-written review of literature about vitamin K prophylaxis and vitamin K deficiency bleeding (VKDB) in newborns, which was utilized to inform standard practice in Belgium.  Survey results about current practices in a maternity ward in Flanders are also included. The authors advised that 1-2 mg given intramuscularly (IM) to term infants and 1 mg IM to pre-term infants (<32 weeks). If parents refuse, 2 mg is recommended orally at birth followed by 1-2 mg/week for 3 months for breast fed infants. These advised amounts are consistent with the scientific literature and practices in other countries.

We thank the reviewer for the nice comments

The authors might consider indicating that multiple forms of vitamin K exist (phylloquinone and menaquinones), how they differ, and that most countries recommend phylloquinone for vitamin K prophylaxis, except Japan which uses menaquinone-4 (Shearer M, Blood, March 2009, Pages 49-59).

The following was added to the manuscript: Different forms of vit K exist: phylloquinone and menaquinones [43]. Phylloquinone, which is the major dietary source, is concentrated in leafy plants and is the vitamin K form best characterized in terms of food composition and dietary intakes. In contrast, menaquinones are the product of bacterial production or conversion from dietary phylloquinone. Most countries recommend phylloquinone for vit K prophylaxis, except Japan which uses menaquinone-4 [43].

Another option would be to include "vitamin K form' as a column in Table 10.

We added the above text to the manuscript.

The authors indicate that lack of gut bacteria in newborns contributes to VKDB (lines 86, 362). However, vitamin K produced by intestinal bacteria is not well-absorbed so its contribution to preventing VKDB is questionable (10.1146/annurev.nu.15.070195.002151). Likewise, while antibiotics can alter the intestinal bacteria, it is uncertain how this affects VKDB (lines 87-88).

We included this comment in the discussion section (l 362). Regarding antibiotics, a comment was added to line 88.

Lines 235-237: It would be helpful to include references so that the reader knows which studies these are.

The respective references were added to lines 239-240 of the revised version.

Round 2

Reviewer 1 Report

Please provide a clean, unmarked copy

including all changes made. Please

check all reference numbers in text and

compare them with reference list at the

end of the manuscript. There are still

minor language and grammar mistakes,

for instance ‘Today, practices varies (it should read ‘vary’).